# A Review on Information Technologies Applicable to Precision Dairy Farming: Focus on Behavior, Health Monitoring, and the Precise Feeding of Dairy Cows

Na Liu [1,2,3], Jingwei Qi [1,2,3,*], Xiaoping An [1,2,3] and Yuan Wang [1,2,3]

1   College of Animal Science, Inner Mongolia Agricultural University, Hohhot 010018, China;
    liuna1988@imau.edu.cn (N.L.); anxiaoping@imau.edu.cn (X.A.); wangyuan@imau.edu.cn (Y.W.)
2   National Center of Technology Innovation for Dairy-Breeding and Production Research Subcenter,
    Hohhot 010018, China
3   Key Laboratory of Smart Animal Husbandry at Universities of Inner Mongolia Autonomous Region,
    Integrated Research Platform of Smart Animal Husbandry at Universities of Inner Mongolia, Inner Mongolia
    Herbivorous Livestock Feed Engineering Technology Research Center, Hohhot 010018, China
*   Correspondence: qijingwei@imau.edu.cn

**Abstract:** Milk production plays an essential role in the global economy. With the development of herds and farming systems, the collection of fine-scale data to enhance efficiency and decision-making on dairy farms still faces challenges. The behavior of animals reflects their physical state and health level. In recent years, the rapid development of the Internet of Things (IoT), artificial intelligence (AI), and computer vision (CV) has made great progress in the research of precision dairy farming. Combining data from image, sound, and movement sensors with algorithms, these methods are conducive to monitoring the behavior, health, and management practices of dairy cows. In this review, we summarize the latest research on contact sensors, vision analysis, and machine-learning technologies applicable to dairy cattle, and we focus on the individual recognition, behavior, and health monitoring of dairy cattle and precise feeding. The utilization of state-of-the-art technologies allows for monitoring behavior in near real-time conditions, detecting cow mastitis in a timely manner, and assessing body conditions and feed intake accurately, which enables the promotion of the health and management level of dairy cows. Although there are limitations in implementing machine vision algorithms in commercial settings, technologies exist today and continue to be developed in order to be hopefully used in future commercial pasture management, which ultimately results in better value for producers.

**Keywords:** information technologies; precision dairy farming; individual recognition; behavioral monitoring





## 1. Introduction

The dairy industry is an efficient animal husbandry industry, and it includes the breeding of herbivorous animals (such as cows, goats, and sheep), the production of raw milk, and the processing and selling of dairy products. Milk production provides a great contribution to the global economy. According to the United Nations' Food and Agriculture Organization, milk production and the price index of dairy products are growing annually. For instance, the unit yield of dairy cows and the overall level of raw milk processing in China have been exhibiting an upward trend from 2014 to 2021 [1]. In 2021, the year-on-year growth of milk output was 7.1 percent, and this has reached a new high in recent years in China [1].

Information technologies for collecting fine-scale data can prompt efficiency and decision-making on dairy farms [2]. It is clear that information technology, such as the Internet of Things (IoT), artificial intelligence (AI), and computer vision (CV), is displaying

potential for the enrichment of livestock management processes within a precision farming setting [3]. As mentioned, IoT can support farmers with wearable sensor devices for acquiring real-time data to examine numerous factors like cow's behavior, health, milk production, and feed consumption [4]. AI's fundamental and ultimate ambition is to develop machine intelligence (MI), using intelligent machines to perceive, reason, learn, discover, optimize, act, communicate, and reflect upon ideas humanly, rationally, and ethically [5]. AI can be used for inferences related to decision-making on farms. It has emerged as a tool that empowers farmers in monitoring, forecasting, and optimizing farm animal growth, pre-clinical disease detection for early intervention, and monitoring farm animals along with farm management, which contributes to the profitable production of raw milk [6,7]. Deep learning (DL) is a branch of AI that can automatically extract features via learning algorithms, and it is applied to complicated computer vision tasks such as detection, classification, recognition, and tracing [8]. Machine vision technology that is mainly based on image and video processing can produce real-time responses and carry out judgments relative to various animal behaviors [9].

Research has demonstrated the possibility of combining data from image, sound, and movement sensors with algorithms for dairy farm decision-making [10–12]. Ding et al. (2022) used a wearable device equipped with accelerometers to measure the feeding behavior of cattle, and fourteen machine-learning models were established and compared in order to predict feed intake rates [13]. Gardenier et al. (2018) presented a perception system that is suitable for automatic lameness detection. Kinect-v2 3D sensors were placed above and alongside cattle exiting an automated milking system, and they recorded the gait at over a length of 3.6 m. They captured dimensional near-infrared images of cows passing through the system and trained a faster R-CNN that provided accurate detections of hooves and carpal/tarsal joints. Hoof detections were projected into 3D space and tracked to form 4D trajectories (space and time) of each of the four hooves, which identified the lameness of the cows [14].

Precision dairy farming is defined as the utilization of information and communication technologies for the improved control of fine-scale animals and physical resource variability to optimize economic, social, and environmental dairy farm performance [15]. Earlier investigative reports from California found that approximately 69% of producers adopt monitored technologies on farms for daily milk yield, cow activity, and mastitis [16], while in Brazilian farms, the adoption of information technologies is still considered low due to economic issues and the farmer's lack of knowledge about the technologies or the importance of the parameters monitored [17]. However, this study points to higher milk yield in farms with higher levels of technology adoption [17]. Similar investigations in Australia showed that farmers with more than 500 cows adopted between two and five times more specific precision technologies, such as automatic cup removers, automatic milk plant wash systems, electronic cow identification systems, and herd management software, when compared with smaller farms [18], which could reduce labor needs.

Several information technologies have been adopted to help breeders collect reliable information on individual cows in their herds and to monitor changes in cow behavior that are indicators of changes in physiological statuses, such as estrus [19]. These technologies were mostly discovered in research and are not yet available on commercial farms. In this review, we summarize the body of research on recent innovations in information technologies that could be implemented in dairy farming. These use cases are focused on the individual recognition, behavior, and health monitoring of dairy cattle and precise feeding, and these exhibit substantial commercial application potential. Consequently, we propose that the implementation of recent innovations in information technologies, such as IoT, AI, CV, and DL, enables the promotion of the health and welfare of the entire life cycle of dairy cattle and the creation of better value for producers.

## 2. Approach

We conducted a mindful scientific literature search. Our goal was to find the latest peer-reviewed papers that not only deal with precision dairy farming but also include behavioral identification, health monitoring, and precise feeding of dairy cattle. The literature search was conducted in four scientific databases: Scopus; PubMed; Science Direct; as well as Web of Science. A systematic approach was adopted to narrow down the search results to papers that are directly related to the scope of precision dairy farming. The initial search started with a broad search equation comprising basic keywords—"precision dairy farming" AND "behavior" AND "health" AND "feeding"—to obtain extended search results. The year of publication was set using a custom range from 2018 to 2023 to ensure the time sensitivity of the topic. The literature was selected based on the criteria mentioned in Figure 1.

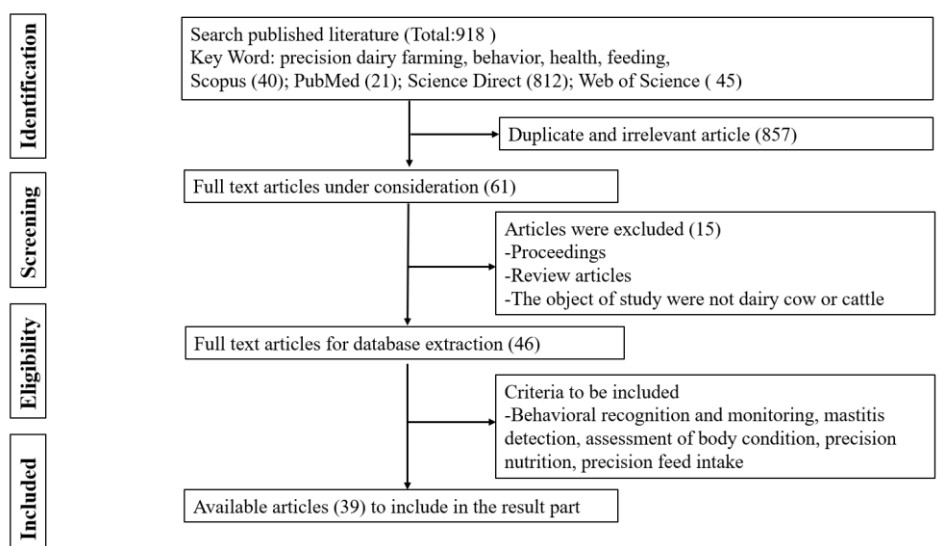

**Figure 1.** Flow chart of scientific literature search and selection for this review.

## 3. Results and Discussion

### 3.1. Individual Recognition of Dairy Cows

With farm scales becoming larger, we need more tools to identify individual animals while having less skilled labor available in a herd setting. The effective and accurate recognition of individual dairy cattle is the prerequisite and foundation for recording and analyzing the animal's behavior automatically [20]. Radio frequency identification (RFID) technology is an accurate, convenient, and rapid method of identifying cattle identities, and it has been widely used in dairy farms [21]. RFID is the typical electronic identification device usually carried out by marking electronic ear tags. When an RFID tag passes through the field of the scanning antenna, it detects the activation signal from the antenna, which "wakes up" the RFID chip, and it transmits information on its microchip that is picked up by the scanning antenna. RFID can also be a tool for dairy managers, and its use can result in the efficient management of large herds via automatic weighting and health monitoring [21]. In dairy farming management, the use of RFID non-contact automatic identification technology can provide accurate individual data for each cow, which is of great significance for the orderly management and monitoring of the entire process of breeding links. Mirmanov et al. (2021) developed automatic cattle weighing systems with RFID systems, and they have passed experimental tests and allow for assessing not only the dynamics in weight changes but also accurately displaying the weight of the animal [22]. However, the RFID device usually needs to be attached to an animal, which may be lost, removed, or damaged [23]. Deep-learning approaches with powerful feature extraction and image representation abilities also have been applied for cattle identification purposes [11]. These comprise a non-contact method of cow identification and exhibit higher recognition accuracy; they are represented by convolutional neural networks and can not only learn

and classify the target in the image but also accurately predict the location of the target [22]. For instance, Kumar et al. (2018) proposed a CNN-based approach to identify individual cattle using primary muzzle point images, and 98.99% accuracy was achieved [24]. Shen et al. (2020) used the you only look once (YOLO) model to detect cow objects and then fine-tuned an AlexNet CNN model to classify each cow. The results showed an accuracy of 96.65% in terms of cow identification [25]. Preliminary computer science research suggests the possible application of DL in the individual recognition of dairy cows. It is not ready for farms yet.

Moreover, there exist some characteristic and stable differences between the vocalizations of individual cows, and some experienced husbandry men can recognize these; this recognition is realized by hearing the cow's voice from a distance [26]. Thus, the individual differences in cattle vocalization can potentially be used as clues to an individual's identity. Jung et al. (2021) developed a deep-learning speech classification model to determine the status of cattle by monitoring the voices of cattle on an experimental farm. The developed model was deployed as a web platform that provides information obtained from a total of 12 sound sensors, providing cattle vocalization monitoring in real time and enabling the researcher to determine the status of their cattle [27]. In addition, D'Urso et al. (2023) designed the SEWIO ultrawide-band (UWB) real-time location system for the identification and localization of cows in barns and in laboratory conditions. It has been reported that the SEWIO UWB system was useful for locating an animal's position in the barn and for measuring time spent in a specific area in the barn [28]. However, it is necessary to verify the usability of vocalizations in identifying individual cows' in-field conditions in future studies.

### 3.2. Behavioral Monitoring of Dairy Cattle

The behavior of the animals reflects their physical state, and monitoring the basic behavior of cows (e.g., food intake, rumination, and walking) might help in evaluating physiological health and treating dairy cow diseases [29,30]. Therefore, the monitoring of behavior is essential for optimizing animal performance, welfare, and timely management decisions [11,31]. Currently, considerable research studies about the monitoring of cow behavior have been carried out, and they have achieved innovative results using information technology [32,33] (Table 1).

### 3.2.1. Research on Behavioral Recognition

The recognition of cow behavior has the potential to decrease manual labor and enhance management efficiency. In recent years, contact sensors (including accelerometers, inertial measurement units (IMU), pedometers, and magnetometers) have usually been designed to collect different behavioral movements and recognize and track animal behaviors [11,19,34–36]. It is reported that wearable behavior-monitoring systems that are integrated with sensors, such as collars, ear tags, and leg bands, have been used to autonomously identify dairy cow behavior while minimizing human interference or human error [12,35,37,38]. Together with the sensors, machine-learning techniques were applied, including various algorithms, such as random forest (RF), decision tree (DT), and K-nearest-neighbors (KNN), to classify the different behaviors of cows [36,38–40]. As shown in Table 1, Shen et al. (2020) used a triaxial acceleration sensor to collect and classify jaw motion data in order to identify dairy cows' ruminating and feeding behaviors using three machine-learning algorithms. The results show that the accuracies of best feeding and ruminating behavior recognition were 92.80% and 93.70%, respectively [41]. Similarly, Balasso et al. (2021) also adopted triaxial acceleration sensors and four algorithms to recognize the posture and resting behavior of dairy cows, and they observed that the best accuracy for predicting posture was 0.99, using the extreme boosting algorithm (XGB), whereas the highest overall accuracy of predicting behaviors was 0.76, using the RF model [36]. Tian et al. (2021) adopted a multi-sensor to collect data on cows' multi-behaviors and recognized seven types of behaviors (feeding, ruminating, running, resting, head shaking, drinking,

and walking) using KNN and RF models. They found that the KNN-RF fusion model had the highest average recognition accuracy of 98.51%, in which the recognition of dairy cow feeding behavior exhibited a recognition accuracy of 99.34% [39]. Beyond that, there are some research studies related to the behavioral monitoring studies of dairy calves. For instance, Carslake et al. (2020) proposed a classification algorithm that was able to accurately identify multiple behaviors in dairy calves using a sensor, such as self-grooming, feeding, resting behaviors, and locomotor play [42]. However, direct contact devices have drawbacks such as high cost and can be easily damaged, which may lead to stress reactions in dairy cows that are not conducive to animal welfare.

In contrast, machine vision technology, as a non-contact, non-stressful method, can recognize cows' behaviors without interrupting them, enabling the monitoring of their basic motion behaviors with higher accuracy and efficiency [43]. Wu et al. (2021) proposed the fusion of VGG CNN and long short-term memory (CNN-LSTM) algorithms in order to instantly and accurately identify the five basic behaviors (drinking, ruminating, walking, standing, and lying) of dairy cows in complex environments that have low-quality surveillance videos, complex illumination, and weather variations. The precision for the recognition of five basic behaviors ranged from 0.958 to 0.995 [32]. Ma et al. (2022) collected a total of 406 videos containing 256,500 frames of dairy cows in different scenes and postures and proposed an algorithm designed for the recognition of cows' basic motion behaviors using Rank eXpansion Network 3D (Rexnet 3D). The proposed method effectively distinguished the lying, standing, and walking behaviors of dairy cows in natural scenes, and the recognition accuracy reached 95.00% [43]. Wei et al. (2023) presented a pose estimation method for cows based on the spatiotemporal features of the skeleton, and they observed that the average precision of the key points (APK) for the pelvis in the standing and lying poses achieved 89.52% and 90.13%, respectively, which validated the effectiveness of skeleton extraction to estimate the pose of cows [44].

### 3.2.2. Research on Behavioral Monitoring

- Feeding behavior

Feed intake is a principal factor that affects the lactation of dairy cows [45]. Abnormal feeding behavior may be related to the illness of dairy cows. Monitoring the changes in the feeding behavior of dairy cows is critical for evaluating their milk production and health status. A selection of research studies applies vision analysis and machine-learning technology to achieve feeding behavior monitoring. As shown in Table 1, Kuan et al. (2019) developed an embedded imaging system for automatically monitoring individual dairy cow's feeding time. The results demonstrated that the prediction of the feeding time of dairy cows obtained by the imaging system was found to be comparable to manual observations with an $R^2$ value of 0.7802 [40]. Similarly, Achour et al. (2020) developed a real-time image analysis system to monitor the feeding behavior of dairy cows. Different classifiers based on the Caffe CNN model were used to analyze images. The results suggested that the system had 92% accuracy in classifying the standing and feeding states of cows [46]. Yu et al. (2022) presented a DenseResNet-you only look once (DRN-YOLO) deep-learning method for monitoring dairy cow feeding behavior. The DRN-YOLO model detected the feeding behavior of cows photographed from the front with precision, recall, mAP, and F1 scores of 97.16%, 96.51%, 96.91%, and 96.83%, respectively, and this addressed the difficulties of existing cow feeding behavior detection algorithm, such as low accuracy and sensitivity to open farm environments [47].

In addition, wearable collection devices are also a viable method for monitoring animal feeding behaviors. Chelotti et al. (2020) proposed an online bottom–up foraging activity recognizer algorithm (BUFAR) based on the recognition of jaw movements using sound, and this had the great advantage of low computational cost (Table 1). The BUFAR, the incorporated multilayer perceptron (MLP), achieved F1 scores that were higher than 0.75 for both grazing and rumination in the 5-minute detection window size, which outperformed a commercial rumination time estimation system [45]. Similarly, Li et al. (2021) revealed

that the technique for combining collected sound data with deep-learning algorithms could monitor dairy cow feeding behaviors (bites, chews, and chew–bites) and forage species (alfalfa vs. tall fescue) and heights (tall and short) significantly influenced the amplitude and duration of the feeding sounds of dairy cows [48]. Although sound sensors have good performance for monitoring chewing behavior, they are susceptible to being affected by noise in complex farms [49]. There is another monitoring method based on acceleration sensors that monitor the feeding behaviors of dairy cows. These devices are usually fixed on the head, mandible, ear, neck, or other parts of cattle; then, they identify the feeding behaviors by distinguishing the movements and postures of the acceleration [50,51]. For instance, Chen et al. (2022) proposed a machine-learning approach that aims to eliminate the influence of the initial pressure of the noseband pressure sensor with respect to the identification of rumination and eating behaviors. The findings revealed that combined with commonly used data-processing algorithms and time-domain feature extraction methods, a recognition accuracy of 0.966 was observed in both rumination and eating behaviors [52]. Currently, a halter-based sensor would have limited applicability for commercial use outside of research settings.

- Estrus behavior

Estrus is a behavioral sign that ensures that female animals are ready to mate when they are close to the time of ovulation [53]. The timely monitoring of estrus information relative to cows is conducive to mating, reducing calving spacing, and improving farm benefits [54]. During estrus, both external behavior and the internal physiological characteristics of dairy cows have apparent changes. External changes are mainly reflected in increased activity and reduced lying time, while internal changes are manifested in increased body temperatures and increased vaginal mucus secretion [55]. There are two main methods for monitoring the estrus behavior of dairy cows: a contact method based on electronic sensors and a non-contact method based on computer vision [56].

In recent years, a number of automated systems utilizing activity sensors (e.g., pedometers, accelerometers, and voice sensors) have been developed to monitor the specific changes in a certain kind of estrus-accompanied behavior. As shown in Table 1, Schweinzer et al. (2019) used a 3D accelerometer integrated into an ear tag (SMARTBOW, Smartbow GmbH, Weibern, Austria) for the detection of estrus events in indoor-housed dairy cows. The result revealed that the sensitivity, specificity, and accuracy of the SMARTBOW system for detecting the estrus events of multiparous cows were 97%, 98%, and 96%, respectively [57]. Again, Wang et al. (2020) developed an automatic data acquisition system to continuously monitor the location and acceleration data of cow activities in estrus [58]. They found that the proposed estrus detector based on machine-learning techniques showed improved performance, an enhanced number of successful alerts, and a reduced number of false positives compared to statistical analysis methods. The results illustrated that the integration of location, acceleration, and machine-learning methods applies to dairy cow estrus detection [58]. There is a review that pointed out that detections based on sensor-supported activity monitoring are the most practical for estrus detection according to current research studies [59]. Additionally, computer vision as a non-contact method can be used for cow estrus monitoring. Wang et al. (2022) captured videos of cow mounting in a natural breeding scene and proposed using an improved YOLOv5 model to detect the estrus behavior of cows in natural scenes and complex environments (Table 1). The results proved that the average accuracy of the improved model was 94.3% and precision was 97.0%, which are both higher than those of mainstream models such as YOLOv5, YOLOv3, and Faster R-CNN [56]. In order to detect estrus in a timely manner and carry out insemination promptly, Lodkaew et al. (2023) designed an automatic estrus detection system (CowXNet) that relied only on a monitoring camera to detect cows in estrus according to various visual estrus behaviors. The system consists of the following four modules: cow detection; body part detection; estrus behavior detection; and behavioral analysis. The result revealed that CowXNet is promising, and the accuracy of estrus behavior detection was 83% [60].

Cows may show variations in their body temperature during the estrous period. Thus, the heat monitoring method based on thermal infrared images is also a non-contact cow estrus monitoring method. For instance, Wang et al. (2023) proposed a lab–color–space-based feature extraction method based on the thermal infrared images of cow eyes and vulvas to monitor cow estrus. LOGISTIC and SVM (support vector machine) models were used to establish the cow estrus model using the thermal infrared temperature variation in cows in estrus and cows not in estrus. The results showed that the heat detection rate of the LOGISTIC-based model was 82.37%, and the heat detection rate of the SVM-based model was 81.42%, using the optimal segmentation profile [61].

The above research studies sufficiently proved the feasibility of information technology in cow behavioral recognition and monitoring, particularly in feeding behavior and estrus behavior. It is worth noting that there are certain limitations with respect to computer science inferences that use machine vision algorithms in commercial settings. Specifically, the computing power required is often not commercially affordable for a producer, and these types of unsupervised deep-learning systems require a host source that is outside the technological ability of a dairy producer (i.e., they often require a computer scientist to manage them); moreover, the topic of overfitting must also be discussed. One of the challenges of complex CNNs is that they require a substantial amount of data to make accurate inferences without overfitting the data. Since estrus events only occur in cattle once every 21 d, it would require a very large and diverse dataset with multiple environments to make a CNN work at the commercial level. Thus, machine vision technology is not widely used in business.

**Table 1.** Research on the behavioral monitoring of dairy cows based on information technologies.

| Author | Year | Type | Approach | Data Sources | Result |
|--------|------|------|----------|--------------|--------|
| Achour et al. [34] | 2019 | Behavioral recognition | DT [1], finite mixture models | IMU [2] | Recognized standing, lying on each side, and the changes between positions. |
| Tamura et al. [35] | 2019 | Behavioral recognition | DT model | Three-axis accelerometers | Recognized three behaviors of cows (including eating, rumination, and lying). |
| Kuan et al. [40] | 2019 | Behavioral monitoring–feeding | MobileNet CNN [3] | Video | The prediction of the feeding time of dairy cows obtained by an imaging system was found to be comparable to manual observation with an $R^2$ value of 0.7802. |
| Shen et al. [41] | 2019 | Behavioral recognition | KNN [4], support vector machine, and probabilistic neural network | A three-axis acceleration sensor | The accuracies of best feeding and ruminating behavior recognition were 92.80% and 93.70%, respectively. |
| Schweinzer et al. [57] | 2019 | Behavioral monitoring–estrus | Algorithms and machine learning | A 3D [5] accelerometer integrated into an ear-tag | The sensitivity, specificity, and accuracy of the SMARTBOW system for detecting estrus events of multiparous cows were 97%, 98%, and 96%, respectively. |
| Carslake et al. [42] | 2020 | Behavioral recognition | AdaBoost ensemble learning algorithm | Sensors | The algorithm was able to accurately identify multiple behaviors in dairy calves. |
| Achour et al. [46] | 2020 | Behavioral monitoring–feeding | Caffe CNN model | Video | The image analysis system had a high-level understanding of the feeding scene. |
| Chelotti et al. [45] | 2020 | Behavioral monitoring–feeding | An online algorithm called bottom–up foraging activity recognizer (BUFAR), multilayer perceptron (MLP), and DT | Sound | The BUFAR-MLP achieved F1 scores that were higher than 0.75 for both grazing and rumination in the 5-minute detection window size, which outperformed a commercial rumination time estimation system. |

**Table 1.** *Cont.*

| Author | Year | Type | Approach | Data Sources | Result |
|---|---|---|---|---|---|
| Wang et al. [58] | 2020 | Behavioral monitoring–estrus | KNN, back-propagation neural network (BPNN), linear discriminant analysis (LDA), and classification and regression tree (CART) | Accelerometer | The integration of location, acceleration, and machine-learning methods can improve dairy cow estrus detection. |
| Balasso et al. [36] | 2021 | Behavioral recognition | RF [6], KNN, XGB [7], and SVM [8] | Triaxial acceleration sensors | The best accuracy for predicting posture was 0.99, using the XGB model, whereas the highest overall accuracy for predicting behaviors was 0.76, using the RF model. |
| Pavlovic et al. [37] | 2021 | Behavioral recognition | A multi-class CNN | Accelerometer collars | Recognized three behavioral states (rumination, eating, and others). |
| Tian et al. [39] | 2021 | Behavioral recognition | KNN, RF models | Multi-sensor | The KNN-RF fusion model had the highest average recognition accuracy of 98.51% in 7 types of cow behaviors. |
| Wu et al. [32] | 2021 | Behavioral recognition | VGG CNN and Bi-LSTM [9] algorithm | Video | The precision for the recognition of five basic behaviors (drinking, ruminating, walking, standing, and lying) ranged from 0.958 to 0.995. |
| Li et al. [48] | 2021 | Behavioral monitoring–feeding | One-dimensional CNN, two-dimensional CNN, LSTM | Sound | The technique for combining collected sound data with deep-learning algorithms could monitor dairy cow ingestion behaviors (bites, chews, and chew–bites). |
| Qiao et al. [30] | 2022 | Behavioral recognition | Convolutional 3D network and convolutional LSTM | Video | Recognized five common behaviors (feeding, exploring, grooming, walking, and standing) of cows. |
| Ma et al. [43] | 2022 | Behavioral recognition | Rank eXpansion Network 3D (Rexnet 3D) | Videos | The proposed method effectively distinguished the lying, standing, and walking behaviors of dairy cows with a recognition accuracy of 95.00% in natural scenes. |
| Yu et al. [47] | 2022 | Behavioral monitoring–feeding | DenseResNet-you only look once (DRN-YOLO) deep-learning method | Images | The DRN-YOLO model detected the feeding behavior of cows photographed from the front with a precision, recall, mAP, and F1 score of 97.16%, 96.51%, 96.91%, and 96.83%, respectively. |
| Chen et al. [52] | 2022 | Behavioral monitoring–feeding | XGB [10] | Noseband pressure sensor | Combined with the commonly used data-processing algorithms and time-domain feature extraction method, a recognition accuracy of 0.966 with respect to both rumination and eating behaviors was obtained. |
| Wang et al. [56] | 2022 | Behavioral monitoring–estrus | Improved YOLO [11] v5 model, K-means clustering | Video | The proposed model can be the fast and accurate detection of cow estrus events in natural scenes and all-weather conditions. |
| Wei et al. [44] | 2023 | Behavioral recognition | Multi-scale temporal convolutional network (MS-TCN) | Images | The average precision of key points (APK) for the pelvis in standing and lying poses achieved 89.52% and 90.13%, respectively. |

**Table 1.** *Cont.*

| Author | Year | Type | Approach | Data Sources | Result |
|--------|------|------|----------|--------------|--------|
| Balasso et al. [62] | 2023 | Behavioral recognition | 8-layer CNN | Tri-axial accelerometer | The precision, sensitivity/recall, and F1 score of a single behavior had the following range: 0.93–0.99. |
| Lodkaew et al. [60] | 2023 | Behavioral monitoring–estrus | YOLO v4, RestNet, DenseNet and EfficientNet CNN | Video | An automatic estrus detection system for cows (CowXNet) is helpful for assisting farmers in detecting estrus cows, and the accuracy was 83%. |
| Wang et al. [61] | 2023 | Behavioral monitoring–estrus | LOGISITC and SVM models | Thermal infrared images | The heat detection rate of the LOGISTIC-based model was 82.37%, and the heat detection rate of the SVM-based model was 81.42% under the optimal segmentation profile. |

[1] DT—decision tree. [2] IMU—inertial measurement units. [3] CNN—convolutional neural networks. [4] KNN—K-nearest-neighbors. [5] 3D—three-dimensional. [6] RF—random forest. [7] XGB—extreme boosting algorithm. [8] SVM—support vector machine. [9] LSTM—long short-term memory. [10] XGB—extreme boosting algorithm. [11] YOLO—you only look once.

### 3.3. Health Monitoring of Dairy Cattle

Diseases are one of the reasons for the decrease in milk production of dairy cows. The good health and well-being of animals are essential to dairy cow farms and the sustainable production of milk [63]. The early detection and handling of cows that are affected by disease are a challenging task, especially in large farms where employees do not have enough time to observe animals and cannot detect first symptoms of diseases. Faruq et al. (2019) developed a dairy cow health management system—combining monitoring systems and detection systems into one application utilizing IoT and intelligent system technology—for health monitoring relative to the detection and handling of cows that have been affected by diseases. They found that the monitoring system can monitor health conditions in dairy cows based on temperature and heart rate with an error rate of 0.6 degrees Celsius and 3.5 beats per minute, while the detection system can diagnose diseases in dairy cows based on physical symptoms with an accuracy rate of 90 percent [64]. The monitoring system is only useful for research settings and is a difficult technology to use in the detection of disease in dairy farms. Likewise, Unold et al. (2020) also presented an automated IoT-based monitoring system, and it comprised hardware devices, a cloud system, an end-user application, and innovative techniques with respect to data measurements and analysis algorithms; the system was designed to monitor the health of dairy cows. It was proven that the system could effectively monitor animal welfare and the estrus cycle in a real-life test [63]. Studies on the health monitoring of dairy cows using information technology are presented in Table 2.

#### 3.3.1. Mastitis Detection

Mastitis is considered one of the most significant diseases of dairy herds, and it affects all areas of the dairy industry, from animal health to lost milk production and lower product quality; moreover, it has significant effects on farm economics [65]. The number of somatic cells in milk, i.e., somatic cell count (SCC), is the most used indicator for assessing udder health statuses in dairy cows. The SCC in milk above 200,000 cell/mL is generally considered abnormal and indicates inflammation in the udder, which is likely to inflict indirect health risks on consumers [65]. Therefore, mastitis detection is essential to sustainable dairy production.

It has been reported that 70 to 80% of mastitis losses were caused by subclinical mastitis [65]. However, detecting subclinical mastitis is challenging due to the absence of any visible indications. Infrared thermography (IRT) is a non-invasive technology that allows the early detection of subclinical mastitis [66], but it is not suitable for automated disease detection on commercial farms since it is biased toward warmer temperatures. As

shown in Table 2, Sathiyabarathi et al. (2018) used forward-looking infrared (FLIR) Quick Report 1.2 software that analyzed the ocular surface temperature (OST) and udder skin surface temperature (USST) of thermographic images for the early detection of subclinical mastitis in indigenous cows. They found that the mean ($\pm$SD) USST of the subclinical mastitis-affected quarter was significantly higher than the body temperature, and an increase in the USST of subclinical mastitis quarters showed a positive linear relation with the SCC with $R^2 > 0.95$ [67]. In the same manner, Zhang et al. (2020) proposed a real-time, lightweight multi-scale target detection algorithm named EFMYOLOv3 (Enhanced Fusion MobileNetV3 you only look once v3), which can be used to detect dairy cow eyes and udders, and they applied the algorithm to the detection of mastitis in dairy cows using thermal infrared images. The results showed that the accuracy of the mastitis classification algorithm was 83.33%, and sensitivity and specificity were 92.31% and 76.47%, respectively [68]. Machado et al. (2021) evaluated the use of thermal imaging carried out via IRT in the detection of subclinical mastitis cases in dairy cows under the commercial conditions of compost barn systems with a semiarid climate [69] (Table 2). They observed that the left fore udder temperature (LFUT, °C), right fore udder temperature (RFUT, °C), rear udder temperature (RUT, °C), and average udder temperature (AUT, °C) were adjusted in quadratic polynomial models with a good prediction of SCC (i.e., infection) with $R^2 = 0.92$, 0.97, 0.86, and 0.94, respectively, which illustrated that IRT is capable of detecting mastitis cases in dairy cows with good precision, especially when using thermal images from the anatomical region of the front quarters of the udder [69]. Wang et al. (2022) used the you only look once v5 (YOLOv5) deep-learning network model to obtain temperature information with respect to the eyes and udders of dairy cows via thermal infrared videos for the detection of mastitis. The detection accuracy of dairy cow mastitis via YOLOv5 and the comprehensive detection method was used to detect cow mastitis with an accuracy of 85.71% [70].

Furthermore, in some automatic milking systems (AMS), fully automated online analysis equipment is available for monitoring the occurrence of mastitis according to the somatic cell count at each milking instance and using a number of additional factors with respect to udder health that are recorded in the system [71]. Norstebo et al. (2019) revealed that the coefficient of variation was 0.11 at the online cell counter (OCC) level relevant for the detection of subclinical mastitis, and a concordance correlation coefficient of 0.82 was obtained when comparing results from the OCC sensor with results from a DHI laboratory [72] (Table 2). Naqvi et al. (2022) developed a recurrent neural network (RNN) model using a diverse range of variables (including milk and behavioral characteristics, cow traits, and farm level/environmental variables) for the detection of clinical mastitis (CM) in AMS farms. They found that SCC, daily variance in milking intervals, and milk temperature were identified as the three most important variables, as defined by their impact on model predictions. The results demonstrated that RNNs can effectively detect over 90% of cases of severe CM by integrating a number of variables that are regularly measured on AMS farms [73]. Milk electrical conductivity (EC) is widely used to detect mastitis in dairy farms. Paudyal et al. (2020) found that characteristic temporal patterns in EC and milk yield (MY), in particular pathogen groups, may provide indications for the differentiation of mastitis, which results in the occurrence of pathogens in Holstein cows [74]. Fan et al. (2023) developed a machine-learning framework to detect and predict clinical mastitis (CM) using imbalanced data recorded by AMS. The result showed that combining the DT-based ensemble models with oversampling techniques achieved relatively high sensitivity (82%) and specificity (95% for CM detection and 93% for CM prediction). Creating models using AMS data from the past seven to nine milkings (approximately 3 d) is recommended for identifying positive CM cases for farmers [75].

Aiming at achieving cost-effective mastitis transmission control, Feng et al. (2021) proposed an IoT-based animal social behavior sensing framework to model mastitis propagation and inferred mastitis infection risks among dairy cows [65]. They used portable GPS devices to monitor cows' social behaviors, and they proposed a flexible probabilistic

disease transmission model to estimate and forecast mastitis infection probabilities. The effectiveness of the framework was demonstrated by both the theoretical and simulation-based analytics of in-the-field experiments. The correctness of the prediction model was also validated by SCC mastitis tests in real-world scenarios.

### 3.3.2. Other Diseases

Diarrhea is the leading cause of dairy calf mortality, and it is characterized by the observation of feces with a loose or watery consistency [76]. These calves also often have reduced activity [77] and compromised milk intake [78] compared with healthy calves. It is reported that individually housed diarrheic calves wearing ear-based accelerometers had longer lying times and reduced activeness compared with healthy calves during the day before and the day after diarrhea diagnosis [77]. Guevara-Mann et al. (2023) conducted case-control studies to quantify the association between daily activity behaviors, relative changes in activity patterns (lying time, lying bouts, step count, and activity index), and diarrhea status in pre-weaned dairy calves. They revealed that diarrheic calves were more lethargic, and they had relative changes in activity patterns—2 d before clinical signs of diarrhea. Specifically, diarrheic calves exhibited fewer steps and had a reduced activity index, and there was an interaction [79].

The metabolic and digestive disorders of dairy cows, such as ketosis, displaced abomasum (DA), and indigestion, can cause losses in milk production and increase treatment costs and the risk of culling and death; these instances may not be helpful with respect to the cow's well-being and farm economic benefit [80]. Stangaferro et al. (2016) suggested that monitoring rumination time and physical activity could be useful for identifying cows with metabolic and digestive disorders in the early postpartum period [80]. Kaufman et al. (2018) determined the associations between rumination time (RT) and health status with respect to milk yield and milk composition. They found that RT was positively associated with milk yield in early lactation dairy cows (4 to 28 d in milk) across all lactation periods, and it was negatively associated with milk fat content in ≥one-third of lactating cows. Furthermore, early lactating cows that experience subclinical ketosis, particularly with one or more other health problems, might have decreased milk yield and milk protein contents [81]. Many sensors have been developed to monitor rumination. Reiter et al. (2018) evaluated the ear-tag-based accelerometer system Smartbow for detecting rumination time, chewing cycles, and rumination bouts in indoor-housed dairy cows. Then, the parameters were determined using the analyses of video recordings as a reference, and they were compared with the results of the accelerometer system. The rumination time, chewing cycles, and rumination bouts detected using Smartbow were highly associated (r > 0.99) with the analyses of video recordings [82].

One of the common diseases of cattle is bovine respiratory disease, and its signs are fever, nasal discharge, and rapid breathing [83]. In such cases, the monitoring of the respiratory rate (RR) is one of the most important indices for animal disease examination. Strutzke et al. (2019) developed a device and mounting hardware for RR measurements in cattle and compared the measured data using a device with the counted RR frequencies of the video recording [84].

Lameness is a major welfare problem on modern dairy farms, and it is associated with physical injury and has more than 40 different clinical conditions that result in reduced milk yield and fertility and increased risks of premature culling and substantial economic loss [85]. Zillner et al. (2018) found that lameness was significantly associated with walking speed. If the cow has previously suffered from lameness for a longer period, then the cow would cover distances at slower speeds [86]. Zhao et al. (2018) analyzed leg swing using computer vision techniques to develop an automatic and continuous system for scoring the locomotion of cows in order to detect and predict lameness. The accuracy of the classification was 90.18%, and the sensitivity and specificity averages were 90.25% and 94.74%, respectively. This research study demonstrates the feasibility of classifying dairy cow lameness based on the six motion features extracted using leg swing analysis. The

results showed that the accuracy of the classification was 90.18%, and the sensitivity and specificity averages were 90.25% and 94.74%, respectively [87].

### 3.3.3. Assessment of Body Conditions

Body condition is a significant welfare and herd management indicator, and it exhibits high correlations with the health and metabolic status of dairy cows [88]. The assessment of body conditions directly impacts the nutritional, health, and reproductive status of dairy herds [89], and it is the most commonly used method for monitoring metabolic diseases, such as ketosis [90]. Acquiring the body condition data of dairy cows every 30 d throughout the lactation cycle is valuable [66]. While the traditional body condition score (BCS) uses manual scoring, it is not encouraged because it is subjective, time-consuming, and stressful for the entire herd [91]. Recently, BCS estimation models based on image analysis and machine-learning techniques have been developed and used to estimate the body conditions of dairy cows [92]. As shown in Table 2, Huang et al. (2019) captured the back-view images of cows using network cameras, manually labeled the key body parts such as tails, pins, and rumps in the images, and then used the single-shot multi-box detector (SSD) method to detect the tail and evaluate BCS. They achieved an accuracy of 98.46% on average for the BCS assessment [88]. Sun et al. (2019) developed an automatic system for identifying individuals and assessing BCS using a deep-learning framework, and online verification was used to evaluate the accuracy and precision of the system. The results showed that the overall accuracy of BCS estimations was high (0.77 and 0.98 within 0.25 and 0.5 units, respectively), and individual identification and BCS assessments performed well in the online measurement, with accuracies of 0.937 and 0.409, respectively [93]. Although the validation for the actual BCS ranging from 3.25 to 3.5 was weak, the system could help production decision-makers reduce the negative energy balance in early lactation via the accurate observation of individuals experiencing rapid declines in body conditions [93]. Similarly, Rodríguez Alvarez et al. (2019) proposed an automatic system to estimate BCS values using transfer learning and ensemble modeling techniques, and the accuracy of BCS estimations within 0.25 units of difference from true values was 82%, while the overall accuracy within 0.50 units was 97% [94]. Mullins et al. (2019) validated the implementation of an automated BCS system in a commercial setting and compared the agreement of automated body condition scores with conventional manual scoring. They found that the automated BCS camera system's accuracy was equivalent to manual scoring, and this may encourage more producers to adopt BCS into their practices in order to detect early signs of BCS changes in individual cattle [95]. Martins et al. (2020) determined the BCS of Holstein heifers and lactating cows using three-dimensional (3D) cameras [89]. The findings revealed that 3D cameras have a good prospective future commercial use; however, the BSC prediction model still requires improvements due to an $R^2$ of 0.63 and 0.61 and RMSE of 0.16 and 0.17 for lateral and dorsal images, respectively [89]. Shi et al. (2023) proposed an automatic scoring method for dairy cow body conditions based on attention-guided 3D point cloud-feature extraction, which achieved accuracies of 0.80 and 0.96 within 0.25- and 0.50-point deviations, respectively. The point cloud classification network with attention guiding has achieved good BCS estimation results in comparison with the other research studies [96].

The accuracy of automated BCS scoring will be improved as machine-learning techniques develop. However, there are some limitations to the machine-learning technology; i.e., it needs to be ready for network connectivity, and data streams often become clogged as data are uploaded to a cloud station in Europe and are redownloaded for interpretation at the farm's base station. Thus, the data have to be cleaned regularly, and there has to be an ethernet connection to an internet source that provides sufficient streaming power to transmit and download the data.

**Table 2.** Research studies on the health monitoring of dairy cows based on information technology.

| Author | Year | Type | Approach | Data Sources | Result |
|---|---|---|---|---|---|
| Sathiyabarathi et al. [67] | 2018 | Mastitis detection | FLIR [1] Quick Report 1.2 software and SPSS [2] 16.0 | Thermographic images | The increase in the USST [3] of subclinical mastitis quarters showed a positive linear relation with an SCC [4] of $R^2$ > 0.95. |
| Norstebo et al. [72] | 2019 | Mastitis detection | A multi-level modeling approach | OCC [5] sensor in automatic milking systems | The coefficient of variation was 0.11 at an OCC level and relevant for the detection of subclinical mastitis, and a concordance correlation coefficient of 0.82 was attained when comparing results from the OCC sensor with results from a DHI laboratory. |
| Huang et al. [88] | 2019 | Assessment of body condition | SSD [6] method | 2D [7] camera | The accuracy of BCS [8] assessments is 98.46% on average. |
| Sun et al. [93] | 2019 | Assessment of body condition | DenseNet CNN [9], stochastic gradient descent algorithm | Image | The overall accuracy of the BCS estimation was high (0.77 and 0.98 within 0.25 and 0.5 units, respectively). |
| Rodríguez Alvarez et al. [94] | 2019 | Assessment of body condition | SqueezeNet CNN, transfer learning, and model ensembling | Image | The overall accuracy of BCS estimations was within 0.25 units of difference from true values up to 82%, while the overall accuracy was within 0.50 units up to 97%. |
| Mullins et al. [95] | 2019 | Assessment of body condition | Algorithm | Commercial automatic BCS camera | The automated BCS camera system's accuracy was equivalent to manual scoring. |
| Zhang et al. [68] | 2020 | Mastitis detection | Enhanced fusion mobileNetV3 YOLO [10] v3 (EFMYOLOv3) deep-learning network | Thermal infrared images | This method can be used for the automatic recognition of dairy cow mastitis. |
| Martins et al. [89] | 2020 | Assessment of body condition | MATLAB [12] R2016b software, GLMSELECT LASSO regression analyses, PROC MIXED of SAS [13] fit the final model | 3D [11] cameras and depth sensor | This model was obtained to predict BCS had an $R^2$ of 0.63 and 0.61 and RMSE [14] of 0.16 and 0.17 for lateral and dorsal images, respectively. |
| Feng et al. [65] | 2021 | Mastitis detection | Data fusion techniques and 4 algorithms | Portable GPS [15] devices | The probabilistic disease transmission model is useful and effective in predicting infected cows. |
| Machado et al. [69] | 2021 | Mastitis detection | Computer program for regression and correlation analyses | Thermal imaging | LFUT [16], RFUT [17], RUT [18], and AUT [19] were adjusted in quadratic polynomial models with good predictions of SCC (i.e., infection) with $R^2$ = 0.92, 0.97, 0.86, and 0.94, respectively. |
| Naqvi et al. [73] | 2022 | Mastitis detection | RNN [20] model | Automated milking systems | RNNs can effectively detect over 90% of cases of severe CM [21] by integrating a number of variables that are regularly measured on AMS [22] farms. |
| Wang et al. [70] | 2022 | Mastitis detection | YOLOv5 deep-learning network model | Thermal infrared video | The detection accuracy of dairy cow mastitis using YOLOv5 and a comprehensive detection method was used to detect cow mastitis at an accuracy of 85.71%. |
| Fan et al. [75] | 2023 | Mastitis detection | DT [23]-based ensemble models | Automated milking systems | Combining the DT-based ensemble models with oversampling techniques achieved relatively high sensitivity (82%) and specificity (95% for CM detection and 93% for CM prediction) |
| Shi et al. [96] | 2023 | Assessment of body condition | An attention-guided 3D point cloud feature-extraction model | Depth image | The point cloud classification network with attention guiding achieved accuracies of 0.80 and 0.96 within 0.25- and 0.50-point deviation, respectively. |

[1] FLIR—forward-looking infrared radar. [2] SPSS—statistical package for the social sciences. [3] USST—udder skin surface temperature. [4] SCC—somatic cell count. [5] OCC—online somatic cell counter. [6] SSD—single shot multi-box detector. [7] 2D—two-dimensional. [8] BCS—body condition score. [9] CNN—convolutional neural networks. [10] YOLO—you only look once. [11] 3D—three-dimensional. [12] MATLAB—matrix laboratory. [13] SAS—statistics analysis system. [14] RMSE—root mean square error. [15] GPS—global positioning system. [16] LFUT—left fore udder temperature. [17] RFUT—right fore udder temperature. [18] RUT—rear udder temperature. [19] AUT—average udder temperature. [20] RNN—recurrent neural network. [21] CM—clinical mastitis. [22] AMS—automatic milking system. [23] DT—decision tree.

### 3.4. Precision Feeding

3.4.1. Precision Nutrition

Obtaining knowledge of the nutrient content of feed ingredients in a cow's diet and carrying out the accurate prediction of the animal's nutrient requirement are certainly big challenges for dairy farmers [97]. Precision animal nutrition and precision feeding are aimed at optimizing the supply and demand of nutrients relative to animals for targeted

performances of animals, characteristics of milk products, as well as economic and environmental outcomes of farms [98]. Research studies on the precision feeding of dairy cows based on information technology are shown in Table 3.

It is reported that the dry matter (DM) concentrations of alfalfa and corn silage in samples collected within farms exhibit large variability in both long-term and short-term periods [99]. In addition, the daily demand for the nutrition of individual lactating cows, such as metabolizable energy (ME), exhibited deviations due to the climate, diet, and animal factors [100]. Duranovich et al. (2021) observed that the deviation of the estimated daily total metabolizable energy requirements of individual cows (MEt) from the actual ME supplied per cow in the herd varied greatly [100] (Table 3). Moreover, many minerals have vital functions in mammals [101]; for example, phosphorus (P) has a role in energy metabolism, whereas copper (Cu) and zinc (Zn) are involved in immune functions [102]. However, over 75% of cows received a ration with excess Co, Cu, Fe, Mn, and Zn in commercial dairy herds compared to National Research Council recommendations, which may lead to detrimental environmental effects [103]. The nutrient concentrations in feedstuffs are characterized by large variabilities, which could modify the nutrient composition of the total mixed ration (TMR) and affect the health and yields of dairy cows, with consequences with respect to farm economic benefit and sustainability [104].

Near-infrared spectroscopy (NIR) technology is a rapid and accurate analytical technique that is used to collect information on the chemical–physical composition of raw materials, TMR, feces, and milk, and it exhibited high potential in predicting the chemical composition of feeds [105]. As shown in Table 3, Piccioli-Cappelli et al. (2019) evaluated the effect of a precision feeding system based on a near-infrared scanner on metabolic conditions and milk yields in lactating dairy cows. The NIR Analyzer was mounted on a scraper of a front miller, and real-time NIR dry matter analysis was performed with respect to each ingredient. They observed that with the system switched on, the deviation of the DM of the target diet and the diets that were really distributed to cows tended to be lower and feed protein utilization exhibited higher efficiencies [104]. Pereira-Crespo et al. (2022) evaluated the predictive ability of NIRS for the estimation of the chemical composition and organic matter digestibility (OMD) of TMR for dairy cows. They revealed that the NIRS prediction models for estimating the OMD of TMR for dairy cows based on chemical parameters showed superior predictive capacity compared to empirical equations [106]. Murphy et al. (2022) developed NIRS calibrations to predict quality parameters, dry matter (DM), and crude protein (CP) in fresh, undried grass on Irish pastures. They found that NIRS can predict DM contents in fresh grass with a high degree of accuracy and CP contents with moderate levels of accuracy, and this has a positive impact on the nutrient requirement estimation for grazing cattle [107].

### 3.4.2. Precision Feed Intake

In the dairy farming industry, more than 60% of farm expenses are devoted to feed [108]. Ensuring the feed amounts for dairy cows and increasing feed efficiency are critical to improving the efficiency of dairy cows. Hence, monitoring feed intake is potentially beneficial for improving farm management decisions and farm milk yield [109]. The assessment of the feed intake of small group samples could be used to reflect the actual feed intake demands of the same type of cows; moreover, it can be used to guide their precise feeding [110]. As shown in Table 3, Bloch et al. (2019) developed a photogrammetry system for evaluating an individual cow's feed intake and tested the accuracy of the best system. The results showed that the feed mass estimation error was 0.483 kg for feed heaps up to 7 kg [111]. The feed measuring system is simple and user-friendly, and it is equipped with inexpensive equipment and cameras, which has the potential for commercial development. Newly developed machine vision and deep-learning models could be used to measure an individual cow's feed intake. Bezen et al. (2020) designed a computer vision system for individual feed intake measurements based on ResNet CNN models and a low-cost RGB-D (red, green, blue plus depth) camera was used. They found that the feed

intake's weight error exhibited an MAE of 0.127 kg and an MSE of 0.034 kg$^2$; moreover, cow identification accuracy was 93.65% in the feeding lane [8]. Likewise, Saar et al. (2022) proposed a real-time machine vision system to predict the individual feed intake of dairy cows. They used a red, green, and blue plus depth (RGBD) camera to acquire feed pile images of two different feed types (lactating cows' feed and heifers' feed), and several models were developed to predict individual feed intake. The results suggested that the transfer learning (TL) models performed best and achieved mean absolute errors (MAE) of 0.12 and 0.13 kg per meal with RMSEs of 0.18 and 0.17 kg per meal for the two different feeds when tested using varied data collected manually in a cowshed. Testing the model with actual meal data that were automatically collected by the system in the cowshed resulted in an MAE of 0.14 kg per meal and an RMSE of 0.19 kg per meal [109].

Furthermore, Shen et al. (2022) proposed a method of optimizing BP neural networks to establish a dairy cow feed intake assessment model, taking the cow's body weight, lying duration, lying times, walking steps, foraging duration, and concentrate–roughage ratio as input variables and taking the actual feed intake as the output variable, and the model was trained and verified using experimental data collected on site. They concluded that the established BP model using the polynomial decay learning rate has the highest assessment accuracy for the assessment of feed intake, and R$^2$ was 0.94 [112]. In addition, Ding et al. (2022) evaluated an integrated machine-learning algorithm framework to identify jaw movements during feeding using a triaxial accelerometer at a relatively low sampling frequency, and it was also used to predict feed intake on the basis of the acceleration variables of ingesting and chewing activities. The results showed that three feeding activities—ingesting, chewing, and ingesting–chewing—could be effectively identified using the XGB and Viterbi algorithms with a precision of 99% [13].

In brief, the usage of near-infrared spectroscopy to balance diets in mixers is one of the most widely used precision technologies in the dairy sector and has revolutionized the way we feed cattle, especially when using handheld NIRS. On the contrary, no dairy farm knows the true feed intake of each cow, and this requires more preliminary research.

**Table 3.** Research studies on the precision feeding of dairy cows based on information technology.

| Author | Year | Type | Approach | Data Sources | Result |
|---|---|---|---|---|---|
| Piccioli-Cappelli et al. [104] | 2019 | Precision nutrition | Shapiro–Wilk test, a mixed model | NIR [1] analyzer | With the system switched on, the deviation of the DM [2] of the target diet and diets distributed to cows tended to exhibit a lower and higher efficiency with respect to feed protein utilization. |
| Bloch et al. [111] | 2019 | Precision feed intake | MATLAB [3], photomodeler scanner | Camera | The feed mass estimation error was 0.483 kg for feed heaps of up to 7 kg. |
| Bezen et al. [8] | 2020 | Precision feed intake | ResNet CNN [4] | Images | The feed intake weight error was an MAE [5] of 0.127 kg, and MSE [6] was 0.034 kg$^2$; cow identification accuracy was 93.65% in the feeding lane. |
| Duranovich et al. [100] | 2021 | Precision nutrition | Linear extrapolation, orthogonal polynomials of third order, regression models | Proximal hyperspectral sensing coupled with a canopy pasture probe system | The deviation of the daily estimated ME$_t$ [7] requirements of a cow from the actual ME [8] supplied per cow in the herd varied greatly. |
| Duplessis et al. [103] | 2021 | Precision nutrition | The computer of the feeding robot | Electronic scale, Lactanet database. | Above 75% of cows received a ration with excess cobalt, cuprum, ferrum, manganese, and zinc; among them, ferrum and cobalt were the most overfed minerals. |

**Table 3.** *Cont.*

| Author | Year | Type | Approach | Data Sources | Result |
|---|---|---|---|---|---|
| Pereira-Crespo et al. [106] | 2022 | Precision nutrition | CENTER algorithm, modified partial least squares regression | Online NIR spectrophotometer | The NIRS prediction models for estimating the OMD [9] of the total mixed ration of dairy cows based on chemical parameters showed superior predictive capacity than empirical equations. |
| Saar et al. [109] | 2022 | Precision feed intake | TL [10] models based on EfficientNet CNNs | Images of feed piles | The TL models performed best and achieved mean absolute errors of 0.12 and 0.13 kg per meal with an RMSE [11] of 0.18 and 0.17 kg per meal for the two different feeds when tested on varied data collected manually in a cowshed. |
| Shen et al. [112] | 2022 | Precision feed intake | SVR [12] model, KNN [13] logistic regression model, traditional BP [14] neural network model, and multilayer BP neural network model | Smart collar device | The established BP model using the polynomial decay learning rate has the highest assessment accuracy for assessing feed intake; $R^2$ is 0.94. |
| Ding et al. [13] | 2022 | Precision feed intake | Extreme gradient boosting, hidden Markov model, Viterbi algorithm (HMM–Viterbi) | Triaxial accelerometer | This method could effectively identify three feeding activities—ingesting, chewing, and ingesting–chewing—with a precision of 99%. |

[1] NIR—near-infrared spectroscopy. [2] DM—dry matter. [3] MATLAB—matrix laboratory. [4] CNN—convolutional neural networks. [5] MAE—mean absolute error. [6] MSE—mean square error. [7] $ME_t$—total metabolizable energy. [8] ME—metabolizable energy. [9] OMD—organic matter digestibility. [10] TL—transfer learning. [11] RMSE—root mean square error. [12] SVR—support vector regression. [13] KNN—K-nearest-neighbors. [14] BP—back propagation.

## 4. Conclusions

This review explicitly discussed the applications of information technology for precision dairy farming. IoT, AI, and CV have been increasingly utilized to monitor the behavior, health, and management practices of dairy cows. Both contact sensors and CV techniques have been shown to be useful and effective for the real-time behavioral and health monitoring of dairy cows. CV techniques enable the identification of individual cows and the monitoring of cow behaviors (including feeding and estrus behavior) with higher accuracy and efficiency. However, there are some limitations that affect the implementation of CV in commercial settings. Specifically, a dairy producer would have to bear expensive computing costs, employ a computer scientist to manage a host source, and collect a substantial amount of data to carry out accurate computer science inferences without overfitting the data. For the health detection section, some IoT-based monitoring systems are proposed to detect diseases, such as mastitis. NIS is one of the most widely used precision technologies used to balance diets in mixers in the dairy sector and has revolutionized the way that we feed cattle. On the contrary, assessing the feed intake of each cow is one of the great challenges for producers. A constantly evolving CV will be helpful in improving the accuracy of feed intake estimation.

In the future, information technologies, including IoT, AI, and CV, for monitoring behavior and health should be carried out in dairy farms. It will help producers of dairy farms understand the health status of cows in real time and improve the smart decision-making level.

**Author Contributions:** Conceptualization, N.L.; methodology, N.L.; validation, X.A. and Y.W.; formal analysis, N.L.; investigation, X.A.; resources, J.Q.; data curation, X.A.; writing—original draft preparation, N.L.; writing—review and editing, N.L.; supervision, X.A.; project administration, J.Q.; funding acquisition, J.Q. and X.A. All authors have read and agreed to the published version of the manuscript.

**Funding:** This research was funded by the National Center of Technology Innovation for Dairy Program (grant number 2022-scientific research-2, 2023-QNRC-10), the Science and Technology Planning Program of Inner Mongolia Autonomous Region (grant number 2022YFHH0072, 2022YFSJ0029), and the Major Science and Technology Program of Inner Mongolia Autonomous Region (grant number 2020ZD0004).

**Institutional Review Board Statement:** Not applicable.

**Data Availability Statement:** Not applicable.

**Conflicts of Interest:** The authors declare no conflict of interest.

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
