# Peer review of "A Review on Information Technologies Applicable to Precision Dairy Farming: Focus on Behavior, Health Monitoring, and the Precise Feeding of Dairy Cows"

_agriculture, doi:10.3390/agriculture13101858_

Round 1

Reviewer 1 Report

This is an an interesting topic. The authors did not address the objectives of this review properly. Please note, I would recommend to rebuilt this review as a systematic reviews according to the PRISMA guidelines. 

I would recommend editing the English language by someone proficient. I am not a native but I have found many editing needs. I am not detailing these as the list would be extensive

Reviewer 2 Report

Dear Authors,

This is a very interesting and useful review of the literature. However, I noticed a lack of information. 

Could you add information about: 

use of PDF for monitoring of acidosis and ketosis

use of milk composition (fat, protein, etc.) for early diagnosis of diseases in dairy cows

use of other milk components, such as LDH, for early diagnosis of mastitis

use of milk progesterone for assessment of reproduction 

information about behavior biomeaker from PDF during heath stress

Information about reticulorum pH for the assessment of cows health

Reviewer 3 Report

This manuscript presents a comprehensive review focused widely on the utilization of state-of-the-art studies, combining information technology with automation and labor-saving research in livestock farming, particularly in dairy cattle. The following comments mainly concern deep learning technologies, which align with my expertise as a reviewer, for the purpose of contributing to the improvement of this manuscript.

1. On line 38, AI and DL are mentioned in parallel, but correctly, DL is a subset of AI.

2. The definition of AI in lines 42-43 lacks proper explication and substantiation. For example, references to well-known academic papers such as doi:10.1109/MIS.2022.3150944 should be used to provide a more precise and well-supported definition.

3. The terminology "CNN," also referred to as ConvNet, serves as a comprehensive descriptor encompassing a foundational structure, which notably includes backbone networks such as ResNet and MobileNet. Unfortunately, this manuscript does not address the distinctions among backbones of various types. Explicitly specifying the backbones utilized in prior studies holds importance due to the fact that the effectiveness and accuracy of CNN are directly linked to the selection of appropriate backbones.

4. The description of "et al." in the body of this manuscript for citing previous studies sometimes includes parentheses with publication years, and sometimes they don't. It would be appropriate to standardize them into a single format.

5. The decimal point in line 438 is not properly formatted.

6. The conclusion in Section 6 seems to lack systematic and comprehensive descriptions, making it insufficient as a survey. The same impression applies to the abstract. Providing a comprehensive summary of the content outlined in Sections 1-5 is essential.

7. The process by which the preceding papers in the references were obtained remains unclear. It's important to specify details such as the search engines used, the range of years searched, keywords entered, etc.

8. While the manuscript includes state-of-the-art papers published in 2023, it may not be exhaustive. For example, the following papers could also be considered for inclusion in this manuscript:

Wei, Y.; Zhang, H.; Gong, C.; Wang, D.; Ye, M.; Jia, Y. Study of Pose Estimation Based on Spatio-Temporal Characteristics of Cow Skeleton. Agriculture 2023, 13, 1535. https://doi.org/10.3390/agriculture13081535

Balasso, P.; Taccioli, C.; Serva, L.; Magrin, L.; Andrighetto, I.; Marchesini, G. Uncovering Patterns in Dairy Cow Behaviour: A Deep Learning Approach with Tri-Axial Accelerometer Data. Animals 2023, 13, 1886. https://doi.org/10.3390/ani13111886

D’Urso, P.R.; Arcidiacono, C.; Pastell, M.; Cascone, G. Assessment of a UWB Real Time Location System for Dairy Cows’ Monitoring. Sensors 2023, 23, 4873. https://doi.org/10.3390/s23104873

Reviewer 4 Report

The objective of this literature review was to focus on the use of precision technologies to manage dairy cattle. The review is very much appreciated, and will benefit the community. However, the review would benefit greatly from clarifying which precision technology systems are commercially available, which are in wide use, and which are in research, preliminary settings. I think this is fundamental for the reader to understand, especially since complex deep learning systems with images, YOLO, and CNN are in the preliminary settings and no system is commercially available on farm yet. I highly recommend going throughout this manuscript to make these distinctions.

Furthermore, the manuscript would benefit greatly from highlighting the limitations of the work presented. For example, everything that is suggested for estrus detection appears awesome, machine learning appears to really improve estrus detection of dairy cattle. However, you need to highlight the limitations of implementing machine vision algorithms in commercial settings. Specifically, that the computing power required is often not commercially affordable for a producer, and that these types of unsupervised deep learning systems require a host source that is outside the technological ability of a dairy producer (i.e., they often require a computer scientist to manage/revise them occasionally), and also the topic of overfitting must be discussed. One of the challenges of complex CNN is that they require a LOT of data to make accurate inferences without overfitting the data. Since estrus events only occur in cattle once every 21 d, it would talk a very large, diverse dataset, with multiple environments to make a CNN work at a commercial level using the same sensor at each facility. You have to highlight the limitations of computer science inference for the lay reader. I would do this for each section to make distinctions of which are farm ready (such as near-infrared spectroscopy which is used on most farms to precision feed), vs. a heart rate monitor (which would never work on a farm because the device is difficult to keep on an animal), vs. something in the middle like body condition scoring which is moving towards DL but is still limited by the demands for connectivity access, and streaming access to make the system work on a commercial farm. 

Table 1. I would add Kaler's work in using a sensor and ML to detect behavioral monitoring in dairy calves. 

Table 2. FLIR IRT tech is only useful in controlled environments, because of its bias in warmer temperature; thus it is not suitable for automated disease detection on commercial farms. I would highlight this in the text. Furthermore, I do not think IRT is widely used on farms and the citation used (62) does not research this concept please remove. I would also include conductivity to detect mastitis as this is used widely in dairy parlors and milk capture is not discussed here. What are the limitations, successes, and benefits of using conductivity to proxy for mastitis detection in cattle?

L43 I would remove creativity here and replace it with "and can be used for inference related to decision making on farm"

L45 I would replace tackling diseases with "pre-clinical disease detection for early intervention" as that is the goal of many AI systems on farms.

L46 I would remove "and" and replace with "the profitable production of raw milk"

L53 I have yet to see a deep learning image processing software deployed on the commercial dairy for farm animal decision making. I would rephrase this to "research has demonstrated the possibility of using deep learning image processing for dairy farm decision making". Camera data is not yet at the level of commercial dairy animal management. 

L60 Be specific, which condition changed in cow hoof's to identify lameness?

L65 I would remove this statement unless you have a citation to support it as it contradicts your statement below about California using a lot of precision technology 70% of farms....

L71 Define the metric for productivity, economies of scale, average milk per cow produced, profitability?

L77 A cow doesn't have variables, I would rephrase to "monitor changes in her behavior that are indicator of changes in physiological status such as estrus."

L90 Is improving milk yield really the challenge on dairies when many farms average 100 lbs per cow? Or is it that farms are getting larger and we need more tools to identify individual animals with less skilled labor available to us in a herd setting? I would rephrase...

L91 I would define how this technology (RFID) works for the lay reader.

L103 I would add the commercially validated partial weight scale system that is used to manage calf weight and growth with automated milk feeders as well.

L103 I think FRID should be RFID

L113 I think it is a bit misleading to imply that YOLO + CNN system is a better replacement for traditional RFID as the amount of data points required, as well as the number of pictures required to deploy such a system is high, and RFID is a plug and play system... I would highlight here that preliminary computer science research suggests the possibility of DL....It isn't ready for farms yet.

L113-115 I don't know of any dairy that uses vocalizations to intervene or identify individual cows. I would remove this, or quantify that it is exploratory in nature.

L132 typo

L155 I would remove that machine vision is cost effective, right now it is not, hence why it is not deployed on any commercial dairy to my knowledge, with the exception of the body condition scoring camera from DeLaval (I would add Mullins et al., 2019) to your body of work.

L171 Include citation

L182 I would highlight limitations of that study as nothing is repeatable at 100% accuracy.

L208 I would add that a halter based sensor would have limited applicability for commercial use outside of research settings.]

L213 Rutting is not a term used for dairy rephase

L217 Add in the myriad of studies or cite a review about the sensors used for estrus detection.

L209-253 Everything that is suggested is wonderful, machine learning appears to really improve estrus detection of dairy cattle. However, you need to highlight the limitations of implementing machine vision algorithms in commercial settings. Specifically, that the computing power required is often not commercially affordable for a producer, and that these types of unsupervised deep learning systems, require a host source that is outside the technological ability of a dairy producer (i.e., they often require a computer scientist to manage them), and also the topic of overfitting must be discussed. One of the challenges of complex CNN is that they require a LOT of data to make accurate inferences without overfitting the data. Since estrus events only occur in cattle once every 21 d, it would talk a very large, diverse dataset, with multiple environments to make a CNN work at a commercial level. You have to highlight the limitations of computer science inference for the lay reader. 

L245 Please highlight the limitations of using infrared imaging for rectal temperature in cattle. I would even add in that this IRT should not replace the other methods such as accelerometers that work in multiple settings. These are highly influenced by ambient temperature, and bias is a problem, thus we cannot use these in commercial environments where the dairy barn temperature changes (see Cantor et al., 2023 in JDS Communications).

L268 Heart monitors work well, but are they commercially viable? I suggest this would be a difficult technology to use to detect disease on dairies, and is useful for research settings only...Please add this in.

For the disease detection section, I would specify you are only going to discuss mastitis and rephrase the section,. Alternatively, you need to add in digestive disorders, metritis, lameness, bovine respiratory disease, and calf diarrhea.

For the body condition scoring section, add in the limitations to the technology (i.e., that it needs ready network connectivity, and that data streams often can get clogged as data uploads to the cloud station in Europe and redownloads for interpretation at the farm base station). Thus, one limitation is it has to be cleaned regularly, and that it has to be ethernet connected to a internet source with enough streaming power to transmit and download the data.

For the precision feeding section, I would highlight that using near infrared spectroscopy to balance diets in mixers is one of the most widely used precision technology in the dairy sector and has revolutionized the way that we feed cattle, especially with handheld NIRS. In the contrary, no dairy knows true feed intake of each cow, and this is more preliminary research. I would make this distinction. 

The manuscript could benefit from a grammatical revision. I didn't want to take up too much space highlighting grammatical issues, but I strongly recommend to the authors to use an editorial service for the following revision. There are many examples, but here are a few where the grammatical structure of the sentence leaves the intention unclear  (L13, L19, L27, L45, L51, L62, L97, L106, L125, L212, L259)

Round 2

Reviewer 1 Report

Dear Authors,

Please see the attachment for the comments.

The quality of English has significantly improved but still minor check is needed.

Reviewer 2 Report

Dear Authors, 

I would like to express my sincere gratitude for your comprehensive and well-structured response to my comments. Your detailed explanations and diligent efforts in addressing each point I raised are highly commendable.

Best regards,

Author Response

Thank you for your reviewing work.

Reviewer 3 Report

The updated manuscript has been appropriately revised in response to the comments raised during the previous review. This is excellent work. As a review paper that provides an overview of the relevant research field, it attains a suitable level of quality and evaluation.

Author Response

Thank you for your reviewing work.

Reviewer 4 Report

Thank you for the thoughtful edits and responses to my suggestions. 

The English has improved considerably. Thank you for these timely edits. 

Author Response

Thank you for your reviewing work.